# Influence of a Rejuvenator on Homogenization of an Asphalt Mixture with Increased Content of Reclaimed Asphalt Pavement in Lowered Technological Temperatures

**DOI:** 10.3390/ma14102567

**Published:** 2021-05-14

**Authors:** Pawel Slabonski, Beata Stankiewicz, Damian Beben

**Affiliations:** 1Institute of Engineering Research Labor Aquila, 42-262 Poczesna, Poland; ps.labor@onet.pl; 2Faculty of Civil Engineering and Architecture, Opole University of Technology, 45-061 Opole, Poland; d.beben@po.edu.pl

**Keywords:** asphalt mixture, rejuvenator, reclaimed asphalt pavement, recycled pavement aggregate

## Abstract

The most technologically advanced form of road construction uses a high content of reclaimed asphalt pavement (RAP) as a component of its asphalt mixture (AM). However, there is a real problem with the effective interaction of RAP and MA. The research herein described presents an effective use of RAP originating from the recycling process of old pavements thanks to the application of an original rejuvenator. Two types of AM were designed concerning the base course of pavement as well as the wearing course and the binder course for various traffic categories. The achieved results show that the rejuvenator improved the homogenization of RAP with the asphalt binder and aggregate in each mixture type. On the basis of the research, the possibility of using paving AM with an increased content of RAP in lowered technological temperatures received a favorable assessment. Mixtures of asphalt concrete containing 40% RAP meet both Polish and German requirements for mixtures intended for heavy traffic pavements. Thanks to use of the rejuvenator, it is possible to compact AM layers containing RAP in a final compaction temperature lowered by about 20 °C. The achieved AM lab test results were confirmed on trial road sections. The rejuvenator used in tested AMs improved the homogenization of RAP with both binder and virgin aggregate. Moreover, the study proved that it is possible to use 20%, 40%, and even 100% RAP contents in the mixtures thanks to the use of the rejuvenator based on plant resin and the creation of conditions enabling the effective homogenization of AM components.

## 1. Introduction

Asphalt, as a binder, is one of the oldest building materials. The appearance of polymers for asphalt modification had great impact on development of road pavement technology. The first worldwide modifications undertaken with the use of the elastomer SBS (thermoplastic elastomer styrene–butadiene–styrene) occurred the 1950s. Modifying asphalt pavements with polymers extends their durability, whereas pavements of ordinary asphalt types do not survive long service and require milling. If a multi-layer road structure is not of proper capacity or does not meet the requirements of a given traffic load category or, if resistance of top layers to rutting or durable deformations is not enough, it is also necessary to exchange the pavement due to progressive degradation [1,2,3].

At present there is a worldwide tendency to search for solutions in construction investment, including roadwork, which make possible the reduction of negative impacts on the natural environment. One such solution is the recycling of asphalt pavements. Technologies taking into consideration aspects of sustainable development are currently highly appreciated in road construction. Within this scope, the effective recycling of existing but destroyed during operation asphalt pavements allows for a complex approach to construction in accordance with the rules of reclamation and reuse of building materials [4,5,6]. Moreover, in cases of material shortage (aggregate, asphalt, cement), and when supplies “pile-up” as is the case at the end of a working season, using reclaimed asphalt pavement (RAP) is a highly desirable and valuable way to increase the efficiency of roadwork investment [7,8].

Across the global asphalt pavement industry, a successive increase in the amount of RAP content used within courses can be observed, exceeding 25% in wearing courses and up to 55% in other courses [9]. In general, contemporary road pavements are 100% recyclable provided they do not contain tar. The most technologically advanced form of pavement recycling uses RAP as an ingredient of asphalt mixtures (AMs) produced in a mixing plant equipped with a special, so-called “black” drum, which serves to heat the RAP. The required quality of the RAP can be reached through the control of its homogeneity. The moisture content of RAP is an important issue [10,11,12,13]. The higher the percentage content of RAP and its moisture content, the higher the temperature required for the aggregate. To reach a proper resultant temperature of the mixture, the aggregate needs to be heated to between 250–300 °C.

West and Copeland (2015) [5] indicate that a high degree of recycling has been reached in Japan, where the average content of RAP in AM amounts at 47%. In addition, the following studies [14,15,16] point at the preferred tendency to increase RAP content in both hot-mix asphalt (HMA) and warm-mix asphalt (WMA). Based on the research works [4,5,6,13], RAP originating from existing roads of classes A, S, GP (the highest road classes in Poland corresponding with highways, express roads, and main motor roads according to the International European road standard) is a high-value ingredient of AM. It has been proved that presence of RAP in AM does not lower the functional parameters of the mixture. For instance, stone mastic asphalt (SMA) with content of 30% RAP constitutes a mixture of a quality matching SMA without RAP [13].

Valdes et al. (2011) [17] present experimental research aimed at characterizing the mechanical properties of AM with a high content of RAP. Two half-dense mixtures of maximum aggregate size 12 and 20 mm underwent assessment, with RAP contents of 40% and 60%, respectively. The results show that large amounts of reclaimed material can be included in AM thanks to proper characteristics and specific technological production regime of mixtures with RAP content.

The study [6] describes AM types containing 20%, 40%, and 50% of RAP. The tests undertaken by Marshall and Duriez were performed on different mixtures with the use of road asphalt and asphalt modified with an SBS polymer. It was observed that parameters of recipe containing recycled asphalt with 20% of RAP are much closer to parameters of the mixtures with the original unmodified road asphalt. Moreover, the addition of the SBS polymer improved the properties of recycled AM even at a high RAP content. Huang et al. (2005) [18] present lab tests with mixing components of mixtures containing RAP. A comparative analysis of a 20% RAP-based pavement created using unmodified AM was performed. The results of this experiment showed that only a small part of the old asphalt in RAP had in fact taken part in the new amalgamation, activation, and progression of the technological process, meaning that other parts created a rigid shell around the RAP. Therefore, there is a real problem of effectively combining RAP with both asphalt binder and virgin aggregate.

The majority of procedures concerning the recycling of HMA use up to a 40% RAP content. Recently, technologies have been developed which allow this content to be increased to 100%, e.g., the All-RAP process, Ammann RAH 100, Rapmaster, Astec RAP King, HyRAP, Alex-Sin Manufacturing, RATech, HERA System, Bagela, RSL, and dense-graded high-RAP [19,20,21,22]. The main barrier against the common usage of 100% RAP is a lack of durability tests of such pavements, as well as the lack of a homogenous and rational system of selection and designing of such mixtures.

On the basis of complex lab tests [23], it has been found that the presence of RAP in AM generated increased stiffness and decreased shear deformation. Stretching resistance was also increased for samples of AM with a higher RAP content. German technical documents such as TL Asphalt-StB 07/13 [24], TL AG-StB 09 [25], and M WA 2009 [26] contain good examples of possible RAP usage.

At present, research and implementation works are being held across the world to replace conventional asphalt liquefiers with solvers originating from plants in order to lower the technological temperature of AM. On the basis of previous experience, it was claimed that the production of ecologically fluxed asphalts and AM in warm technology with an RAP component is possible thanks to the application of liquefiers of vegetal origins.

Zaumanis et al. (2014) [27] include an analysis of lab samples from 100% recycled HMA modified with use of five successively added additives of different types (waste plant oil, vegetable waste, smear, organic oil, and distilled tall oil) and a proprietary rejuvenator (aromatic extract) at levels of 12%. The properties of the binder and the mixture were tested. All the samples showed perfect resistance to rutting, ensuring, at the same time, a larger endurance limit in comparison with the original characteristics of the mixture and the most lowered critical temperature of cracking. Similar tests were conducted by Guduru et al. (2021) [28], where five different rejuvenators (waste engine oil, waste lubrication engine oil (commercial product), tallow, waste vegetable oil, and crude tall oil) were used. The authors suggested that the rutting of AM due to the excessive softening of the rejuvenated bitumen should not be a big problem if the target dosage of the rejuvenator is determined based on the softening point value or viscosity (at 60 °C) test results. Meanwhile, Baqersad and Ali (2021) [29] have investigated the use of a nanomodified asphalt binder in the recycling of RAP material to increase the incorporation of RAP percentages in recycled asphalt. In general, there are many different studies on the use of rejuvenators for the homogenization of asphalt mixtures with RAP [30,31,32,33,34,35,36,37,38,39,40,41,42,43].

Meroni et al. (2020) [44] show that AM containing up to a 30% content RAP can be designed without a rejuvenator. However, a 45% RAP content in a mixture would not be possible without a proper rejuvenator which eases the adhesion of recycled aggregates to asphalt and aggregates.

It can be concluded that using rejuvenators in mixtures of high RAP content will become a common practice. They are being added to RAP-containing mixtures in order to decrease the stiffness of RAP binders and to improve their low-temperature characteristics, making possible to include into the mixture larger amounts of RAP [45].

In light of the studied literature, the subject of the homogenization of RAP with an asphalt binder and virgin aggregate with use of a rejuvenator is important for determining homogenization mechanisms and co-existing phenomena. Thus far, they have not been recognized or described to a satisfactory degree. Proving that the original rejuvenator generates an improvement in homogenization process definitely adds value on the subject of AM containing RAP. Moreover, the literature lacks a comprehensive description of any research methodology of new types of organic rejuvenators. There are no procedures that allow for the evaluation of specific adhesive properties of oil substances that rejuvenate and revitalize asphalt with an RAP content. Additionally, the subject area surrounding the use of rejuvenators in mixtures of increased stiffness is of pioneering character. Research works covering topics such as water and frost resistance and resistance to durable deformations were executed for the designed AM types with addition of RAP and a rejuvenator. On the basis of the lab and field tests on test road sections, an evaluation was undertaken regarding the possibility of to applying an AM with an increased RAP content in lowered technological temperatures through the usage of an authorial rejuvenator. Tests checked to see if the rejuvenator used was responsible for the better homogenization of RAP containing asphalt binder and virgin aggregate. Lab test results were confirmed on test road sections.

## 2. Materials Used in Lab Tests

### 2.1. Characteristics of the Rejuvenator

The new generation of binders constitute mixtures of asphalt and vegetable oil or its derivatives in the form of methyl esters [10]. During the execution of the process, it is possible that their activation determines the usability of these vegetal resources as components of asphalts.

Generally, vegetal components of this type can be used to liquidate both unmodified asphalts and ones modified with polymers. Enriched binders can be used in the following technologies:during warm-mix pavement production with lowered AM production temperatures, below 150 °C,for AM types produced with emulsions or fluxed asphalt (such a binder ensures the workability of AM within a certain time period).

Previous experience in rejuvenator use for warm-mix asphalt types made it possible to formulate the following advantages of the binders:the elimination of vapors causing discomfort in the process of AM paving,the better homogenization of RAP with the asphalt binder and aggregate,the easier paving and compacting of AM in the process of construction,the use of renewable resources.

Considering the above, an original rejuvenator ^®^Asfix Alfa has been created, which can be used as additive to AM type road asphalts, e.g., SMA, porous asphalt, asphalt concrete, etc. It is to be added immediately to the AM production process. It mixes easily with asphalt with a flash point exceeding 210 °C, and its viscosity in 30 °C is about 31 mPas. This rejuvenator is a liquid of light-yellow color with main active components of natural origins. It is intended for all types of hot paved AM types produced with use of ordinary and modified (with thermoplastic elastomers) road asphalts. It can be used as a component of AM types containing RAP. Such actions cause the revitalization of recycled asphalt contained within the RAP through the completion of the old oil fraction in the asphalt. The use of the rejuvenator aims at majorly improving the adhesion quality of the asphalt and aggregate and various degree of small particle pollution together with considerable improvement in workability. This agent improves adhesion in the production process of asphalt mixtures based on RAP, making it possible to increase the content of reclaimed materials in AM. It unifies RAP parts of similar parameters, allowing for the full control of void content, the optimization of content, and the considerable improvement in water and frost resistance of AM types, especially ones with an RAP content. The discussed rejuvenator causes an effective decrease in arduous smell at production, transport, and paving of AM in comparison to other agents which smell bad. Other existing adhesive and liquidating agents are based on hazardous waste and are dangerous substances themselves which generate toxic waste. The discussed rejuvenator is defined as safe and fully biodegradable and made of renewable resources. Asfix Alfa^®^ is a patented component based on a plant recipe with multidirectional advantages facilitating the final benefits of the target mixture through the improvement of the homogenization process. Homogenization is especially important when combining RAP with virgin aggregates and asphalt.

A high flash point makes it possible for the discussed rejuvenator to be used in mixtures of high production temperatures, i.e., with the technology of cold RAP dosing. It does not cause a loss of its own characteristics in comparison with traditional chemical agents which easily undergo destruction and are unstable. The specific physical and chemical properties of the original Asfix Alfa rejuvenator are presented in Table 1. The Asfix Alfa properties are compared with the traditional chemical agents (Wetfix BE and Wetfix AP17, Akzo Nobel, Stenungsund, Sweden).

The research carried out so far shows that adding Asfix Alfa does not significantly change the basic rheological properties of the asphalt when added in contents of between 0.3% and 0.8%. The change in asphalt penetration after adding the rejuvenator, compared to the starting asphalt, is in the range of −10 to +6 (dimensionless unit). Moreover, both the softening point and the temperature of asphalt brittleness with the addition of Asfix Alfa do not change significantly.

### 2.2. Description of AM Content with the Addition of RAP and a Rejuvenator

Research was held on an AM of asphalt concrete of continuous type of grain size 0/22 mm intended for base course under traffic load categories KR3–4 [48]. It corresponds with the designed traffic category related to the classification of design traffic as regards the total number of equivalent standard axes 100 kN in all the design period N100 (expressed in millions of axes falling on a conventional lane). In addition, it amounts at 0.5 < N_100_ ≤ 2.50 for KR3 (i.e., medium traffic load on the calculated road lane) and 2.5 < N_100_ ≤ 7.30 for KR4 (i.e., heavy traffic). The AM content falls within the scope described in design guidelines for road structures [48], which directly match German guidelines [24,25,26]. The examined mixture contained: road asphalt 50/70, limestone filler, and broken dolomitic of various grain sizes. The mixture was enriched in RAP of two fractions, 0/8 mm and 8/16 mm, originating from the compound milling of wearing and binding courses. The determination of the grain size of the AM was carried out in accordance with PN-EN 12697-2 + A1:2019-12 [49]. The test consisted in determining the percentage content of individual fractions of the AM after extracting the soluble binder by sieving on a set of laboratory sieves and weighing the content remaining on each sieve. A set of laboratory sieves with square meshes with an automatic Multiserw Morek (Multiserw Morek, Marcyporeba, Poland) shaker, a forced air dryer (Multiserw Morek, Marcyporeba, Poland), and an RADWAG (Radom, Poland) electronic non-automatic scale with a weighing range of up to 6000 g were used for the tests. Figure 1 presents the grain size curves of virgin aggregates (dolomitic) and RAP as recycled pavement aggregates used in the examined mixtures.

The examined mixture contained asphalt, with the addition of asphalt originating from RAP (5.8% of 0/8 mm grain size and 4.3% of 8/16 mm grain size) and fresh asphalt 35/50.

Designs of two asphalt concrete mixtures were prepared for base course Asphalt Concrete 22P (AC 22P) with 20% and 40% RAP contents, called AM-20% and AM-40%, respectively. The total amount of asphalt in both mixtures was constant and it amounted to 3.9%. The main binder properties are presented in Table 2. Both the mixtures had similar designed grain size curves (Figure 2) in order to compare them, despite containing different amounts of destruction material. Table 3 presents the composition of both of the designed mixtures.

Rejuvenator content was selected in relation to the total content of binder in the AM (Table 4) at a level of 1.3%.

## 3. Research Methodology

### 3.1. General Notes

This study concerns the technology of AM recycling in a mixing plant by means of a hot-mix dosing method of recycled pavement aggregate. Dosing is executed directly to the mixer of plant form in an additional drum where the RAP is heated, whereas in the cold-mix dosing method the RAP goes directly into the mixer.

For the designed AM types containing RAP and a rejuvenator test of volume, resistance to water and frost (based on the indirect tensile strength ratio (ITSR)) were done, in accordance with procedures [48,50,51], as well as tests of their resistance to durable deformations [52].

Test samples were prepared from a mixture produced in lab conditions. Each mixture was produced under standard temperatures (relevant for the hot-mix technology). Prior to dosing, the aggregate was being heated to 210 °C, and the asphalt binder to 165 °C. The recycled pavement aggregate was dosed without heating to temperatures of about 25 °C. To make sure the results were comparable, the recycled aggregate was dried to reach a maximum humidity of about 0.5%. Following production, the AMs were left standing for one hour to reach the temperatures of 135 °C and 110 °C, respectively. Then the samples were compacted in a Marshall Compaction test machine and in an asphalt roller compactor.

### 3.2. ITSR Test

The resistance of the AM samples to water and frost was determined in accordance with [48]. The ITSR was determined on the basis of an indirect tensile strength test held in accordance with PN-EN 12697-23:2017-12 [51] and PN-EN 12697-30:2019-01 [53] and conditioned in two ways:-“dry set”—a set of samples stored at room temperature,-“wet set”—a set of samples conditioned in water at 40 °C, frozen, and then conditioned at 25 °C. Test samples of 100 mm in diameter of lab-produced AM were compacted in a Marshall Compaction test machine (Mulitserw Morek, Marcyporeba, Poland) using 35 hits on each side of the sample.

The test result is the percentage ratio of the average indirect tensile strength of the samples from the “wet set” and the “dry set”. The test used a thermostatic water bath (Multiserw Mark, Marcyporeba, Poland), a forced air dryer, a FRÖWAG (Oberslum, Germany) vacuum apparatus, a HUMBOLDT (Elgin, IL, USA) strength test press with a measuring range up to 50 kN, an electronic RADWAG (Radom, Poland) non-automatic scales with a weighing range of up to 6000 g, and an automatic TOROPOL (Warsaw, Poland) frost resistance test chamber that maintains a temperature of −18 ± 3 °C.

### 3.3. Determination of Resistance to Durable Deformation

The test of the resistance to durable deformation of the AMs was held in accordance with procedure B in air: in a small wheel tracker, in accordance with norm PN-EN 12697-22:2020-07 [52]. Resistance to rutting was determined on two slabs of dimensions 260 × 320 × 60 mm prepared in lab conditions with a rolling device. The test consisted of the assessment of the susceptibility of the AMs to deformation by measuring the depth of the rut formed by repeated passes of a loaded wheel at a set temperature (60 °C) for 10,000 load cycles. The tests were performed with the use of a small CONTROLS (Milan, Italy) wheel tracker, a rolling device (MATEST (Alcore, Italy) compactor for the preparation of plate samples in accordance with PN-EN 12697-33:2019-03 [54]), a forced air dryer, and an electronic RADWAG non-automatic scale with a weighing range of up to 30,000 g.

### 3.4. Determination of the Bulk Density of the AM Samples

The bulk density of the samples taken from the pavement (or prepared by tamping in the laboratory) was determined in accordance with method B described in PN-EN 12697-6: 2020-07 [55]. The test was performed with the use of a thermostatic water bath, a TERMIO (Termoprodukt, Bielawa, Poland) electric thermometer with a reading accuracy of 0.01 °C, an electronic RADWAG non-automatic scales with a weighing range up to 6000 g with the possibility of hydrostatic weighing, an automatic Multiserw Morek rammer (Marcyporeba, Poland), and an analog caliper with a measuring range of up to 500 mm.

### 3.5. Determination of the AM Density

The AM density was determined in accordance with method A described in PN-EN 12697-6:2020-07 [55]. In this case, the test consisted of measuring the volume of the water displaced by the sample placed in the pycnometer. The test used an electronic non-automatic scale with a weighing range of up to 6000 g, a pycnometric water bath (Multiserw Morek, Marcyporeba, Poland), a TERMIO electric thermometer with an accuracy of 0.01 °C, a FRÖWAG vacuum set that maintains the pressure of max 4 kPa, and Duran Schott (Mainz, Germany) calibrated pycnometers.

### 3.6. Determination of the Voids in AM Samples

The voids content in the AM samples was based on the determination of the volume of air voids in the tested sample. The test was performed in accordance with PN-EN 12697-8:2019-01 [56]. The determination consisted of measuring the density and bulk density and on this basis the void content was determined.

## 4. Test Results

### 4.1. Analysis of Volumetric Properties

Table 5 shows the test results of the physical properties of two AM types containing both rejuvenator and RAP (AM-20% and AM-40%). Both mixtures were compacted using hot-mix technology (135 °C).

Based on the tests analysis it was stated that the requirements had been met [48,57,58] within the scope of the volumetric properties of all AMs of grain size up to 22 mm for base course, for traffic categories KR1–2 and KR3–4. The designed mixes AM-20% and AM-40% are characterized by both twin density and bulk density and a similar content of voids. However, the RAP content of 20% in AM creates a voids content bigger by 0.7% than in the case of the 40% RAP content AM. This is indirectly caused by the increased amount of fine fractions (0.063–2.0 mm) contained in the RAP, causing these voids to be better filled.

### 4.2. Water and Frost Resistance Test Results Analysis

The analysis of the water and frost resistance test results of AM (with RAP and rejuvenator) shows that both mixes are resistant to the agents. An increase in RAP content in an AM up to 40% does not cause a decrease in frost resistance. Tests have proved that resistance to water and frost (ITSR) for AM-20% amounts to 92%, and for AM-40% this equals 93%.

On the basis of test results analysis it has also been stated that the water and frost resistance requirements determined in [48] are met in the case of the AMs containing 20% and 40% RAP. Both the mixtures are characterized by very good water and frost resistance and they even meet requirements for wearing courses where the ITSR is at a minimum of 90%.

### 4.3. Analysis of Resistance to Durable Deformations

Table 6 shows the tests results of resistance to durable deformations of the designed AM-20% and AM-40%.

On the basis of the obtained test results analysis, it was found that the requirements determined in [48] and concerning resistance to durable deformations of AM of grain sizes of up to 22 mm had been met for traffic categories KR3–4 and KR5. The rutting resistance results showed that AM-20% was a more advantageous design than AM-40%. The mixture with a greater amount of RAP demonstrated a slightly bigger proportional rutting depth which can indicate the impact of the adhesive agent and its stabilization in the case of rigid mixtures. A lot of mixtures with a large RAP content have a tendency to stiffen and crack. In this case, a balanced value of PRD_AIR_ was obtained, falling within stable limits of rutting resistance.

### 4.4. Compaction Assessment of AM in Lowered Temperatures

The compaction assessment was completed on the basis of voids content determined for AM compacted in a Marshall rammer at two different temperatures, 135 °C and 110 °C.

The analysis of obtained results of compaction in two different temperatures of AM containing the rejuvenator showed that lowering the standard hot-mix technology temperature (135 °C) by 25 °C caused a slight increase in voids content by about 1% (Figure 3). However, it needs to be underlined that for each compaction temperature value and RAP content, the obtained voids content fell within the limits denoted by the technical requirements in [48] (Figure 3).

When the water and frost resistance (ITSR) test results are compared for different compaction temperature values (Figure 4), it can be stated that lowering the temperature by 25 °C in relation to the standard temperature used in hot-mix technology does not cause a considerable lowering of the ITSR index. Moreover, despite an increased content of voids in the AM compacted in 110 °C in comparison with the AM compacted in standard temperature of 135 °C, the ITSR index does not show any noticeable changes at the lowered compaction temperature. This proves the possibility of using lower technological temperatures.

## 5. Road Section Tests

### 5.1. General Remarks

In order to verify and confirm the obtained lab results, we decided to test sections of road structures with base courses of AM-20% and AM-40% which had previously undergone lab tests. One trial section was executed with pavement containing 100% RAP. The AM-100% composition was also based on original, authorial design. Trial sections were executed across the Silesian Voivodship near Czestochowa city (southern Poland). The core samples were cut out of the trial road section using an automatic CEDIMA (Celle, Germany) BW-30 drilling rig equipped with a core drill of the desired diameter, in accordance with PN-EN 12697-27:2017-07 [59]. The drilled cores were removed with forceps and placed in transportation boxes. The determination of the compaction index of the core sample layer cut from the pavement consisted of the determination of the percentage ratio of the bulk density of the sample cut from the trial road section to the bulk density of the reference sample (determined in the laboratory in accordance with PN-EN 13108-20:2008 [60]). Mixtures were designed and tests were executed in a certified laboratory at the Institute of Engineering Research Labor Aquila (Nowa Wies, Poland).

### 5.2. Trial Road Sections for AM-20% and AM-40%

Asphalt concrete pavement was tested on 50–100 m long sections using AM types with RAP contents of 20% to 40% prepared in a lowered compaction temperature of 110 °C. Their production process took place in a mixing plant with a double, so-called “black” drum. Reclaimed asphalt of two fractions, 0/8 mm and 0/16 mm, was used with the addition the rejuvenator Asfix Alfa at a 1.3% content level.

The technology of the AM production was analogous to the one with 100% RAP (item 5.3), the only difference being the fact that the aggregate was dosed separately (it was first heated in a standard drum up to 150 °C). Asphalt and filler (limestone filler) were dosed too. All the virgin materials and the recycled pavement aggregates underwent a mixing process in cycle mixer of the mixing plant. Standard transportation vehicles delivered ready-made mixture up to distances of 15 km.

Figure 5a,b shows example cores of 100 mm diameter extracted from the road structures of AM-20% and AM-40% respectively, composed of wearing course AC 11S, binder course AC 16W, and base course AC 22P. The samples were taken by means of drilling with a crown drill with diamond elements. The samples underwent lab tests.

The checking of the lab tests for each road section was performed in four series, in each of which six normal samples of 100 mm diameter were tested. Figure 6 contains the lab test results for the AM-20% and AM-40% samples, the voids content, and the compaction index for the compaction temperature of 110 °C. The obtained coefficients of variation (0.34–0.48% for bulk density, 0.24–0.46% for pavement density defined by means of volumetric method) indicate the large homogeneity of the physical properties of the pavement types, such as density and bulk density, as regards aspects of voids content and compaction index.

According to the results of tests which were supposed to verify the technical parameters of the mixtures inbuilt in the road pavement, the obtained compaction index was correct; at a value of 98% in both cases (AM-20% and AM-40%) and with voids contents not exceeding 7%. This confirms that the initial designs were correct in their calculations regarding the compliance of both AM-20% and AM-40% with the requirements [56], as well as confirming the correct execution of the road pavement with a proper technological regime.

### 5.3. Trial Road Section with Pavement of AM-100%

As described in the introduction, an AM-100% recipe was developed for the sake of comparison and paved along a 150 m trial section. This asphalt concrete pavement on the trial road section was prepared using remixing technology, i.e., it contained 100% RAP of relevantly selected fractions 0/16 mm and 0/8 mm, with the addition of the rejuvenator Asfix Alfa. The trial road section was divided into four parts, each of which differed in the percentage content of the rejuvenator (0%, 0.7%, 1.0%, and 1.3%) in proportion to the asphalt content in granulate. This trial asphalt pavement was made in the Silesian Voivodship near Czestochowa city. The process of core sample drilling can be seen in Figure 7. The lab tests of the road sections were executed in four series; in each of them six normative core samples of 100 mm diameter were taken.

At the production stage, the RAP was heated to 120–140 °C in a mixing plant with double drums, where the lower drum was supplied with virgin aggregate and RAP and the top one was intended for RAP exclusively.

Lab tests determined the composition of the AM used for the wearing course (Figure 8) and the results were compared to the ones from the asphalt concrete AC 11S contained in [48]. Figure 9a presents a microscopic surface image with visible grains, whereas Figure 9b shows the visible surface crushing of a granulate sample. Previously, RAP of two fractions, 0/16 mm and 0/8 mm, was observed through the use of the microscope KEYENCE VHX 6000 (Keyence Corporation, Osaka, Japan), (Figure 9). The outer surface of the ready AM-100% sample was also analyzed under a microscope (Figure 10). On the basis of the microscope analysis, it can be seen clearly that the rejuvenator has started the rejuvenation and bonding process, and is the factor responsible for the efficient process of homogenization.

Samples were taken from the pavement of the trial road section and their contents of binder and filler were determined. However, prior to this, the determination of the grain size of the extraction was executed. For samples excavated from the pavement (wearing course), both voids content and compaction index were determined (Figure 11). Using the rejuvenator in quantities from 0.7 to 1.3% rendered meeting the requirements impossible [61] with regards to compaction index (>98%) and void content in samples (within the scope of 1.0–4.5%). If the rejuvenator was not used (Figure 11), the compaction index met the requirements [57], however the voids content considerably exceeded the norm (5%). The rejuvenator made the use of 100% contents of RAP in this AM recipe possible. Without it, the homogenization of the mixture components would not be possible.

The obtained coefficients of variation relating to density (0.19–0.55%) and bulk density (0.19–0.38%) showed low values, meaning the pavement displayed high levels of uniformity. However, a double increase in the variation of the sample series without the rejuvenator was observed when compared to the other series containing the agent. This indicates the considerably lower uniformity of executed layer without the rejuvenator.

The content of soluble binder was determined at the level of 4.7% at acceptable value of B_min_ 5.4% [62], and grain composition fell within the boundary points and was in accordance with acceptable deviations. The series of samples without the rejuvenator showed excessive content of voids (9.8%).

Table 7 contains the most important average values from the lab tests for the sample series taken from the test pavements. The obtained results indicate that the test road sections have wearing courses based on granulate, the parameters of which match the ones of asphalt concrete AC 11 S for road category KR1–2. The aggregate composition of the tested mixture falls within the boundary curves recommended in specifications [61]. Moreover, it was stated that the deviations observed did not exceed acceptable ones [63], which proves that the contract conditions had been met by the mixing plant.

## 6. Discussion

Control and acceptance checks of AMs containing RAP are the same as in the case of mixtures made from virgin materials only. In the design of AM with RAP, the lab recipe does not differ from in the design of AMs from new components. One needs to consider all the material components of reclaimed asphalt. Additionally, the parameters of the resultant product need to be checked.

The effective usage of RAP coming from recycled materials was possible thanks to application of a rejuvenator which rejuvenated the asphalt contained in the RAP. Designs of authorial AMs concerned both base courses for traffic categories KR3 and KR4 and wearing and binder courses for KR1–2. The rejuvenator improved the homogenization of RAP with asphalt binder and virgin aggregate in each mixture type.

RAP dosed in cold-mix technology directly to the mixer required more aggregate heating in accordance with [48]. Using high contents of RAP from 20% to 40% together with the rejuvenator requires RAP of a maximum humidity of 2%. The rejuvenator helps towards the effective mixing of the virgin asphalt binder and RAP. However, it is important to follow the recommendations concerning moisture contents in RAP due to the fact that the rejuvenator does not eliminate the negative impact of humidity coming from the RAP and does not neutralize it.

At present, according to [48] it is acceptable to use RAP quantities which do not exceed 20% of AM mass. However, as the research shows, if a rejuvenator is used in cold-mix technology, the RAP content can be increased to up to 40% of the AM mass. Using RAP in quantities exceeding Polish requirements for AM does not contradict European norms, where this practice is fairly acceptable.

Based on the literature, e.g., Meroni et al. (2020) [44], AM containing up to a 30% content of RAP can be designed without a rejuvenator. This study also states that a 45% content of RAP requires the use of a rejuvenator which eases the adhesion of recycled aggregates to asphalt and virgin aggregates. However, the use of Asfix Alfa in each configuration of the RAP content (even 100%) makes it possible to reduce the technological temperature of AM production.

The analysis of the obtained test results for various compaction temperatures showed that lowering the temperature by 20 °C in relation to the standard temperature used in hot-mix technology with the use of fluxed asphalt does not cause a decrease in water or frost resistance. However, lowering the compaction temperature by 40 °C does cause a decrease in the ITSR index by about 10% in comparison to the mixture compacted at 150 °C. These results correspond with voids content in tested mixtures, hence the lack of considerable ITSR change in 130 °C and a drop of ITSR for the compaction temperature of 110 °C, which corresponds with an increased content of voids in this mixture.

Due to protection of limited resources of RAP and binders, as well as the necessity to manage destructed asphalt, it is purposeful to implement solutions effectively using reclaimed materials. At the same time, using recycled materials cannot worsen the properties of AMs produced in this manner. Research results show that producing AMs containing 100% RAP is possible with use of a rejuvenator that helps towards the homogenization of mixture components. It has been proven that, as proportion of rejuvenator in AM-100%, changes in the following properties take place:the bulk density of the samples increases—which proves the increased compatibility of the mixture,the density of the AM decreases,the voids content considerably decreases in comparison to stable increase of the compaction index.

Moreover, it was observed that the increased use of a rejuvenator causes a decrease in voids in comparison to samples where it was not used.

The AM with no additional rejuvenator had an increased number of voids (9.8%) at a compaction index of 99.7%. Using the rejuvenator allowed for the repeated optimal use of the RAP and attained the proper voids content in the pavement, in compliance with requirements [24,25,26,48,61].

Technology using asphalt and the rejuvenator allows for the production and implementation of AMs with RAP which have properties comparable to the ones without granulate. On the basis of the lab tests results it should be stated that AM-40% meets the requirements determined in [11] for mixtures intended for pavements under traffic loads KR1–4 (from light to heavy traffic). The rejuvenator used makes it possible to undertake effective compaction of layers of AM containing RAP in a final compaction temperatures that is lowered by about 20–30 °C. The research works have proved that the original, bio-rejuvenator was responsible for the effective homogenization of the RAP with asphalt binder and virgin aggregate.

Generally, chemical rejuvenators are applied in high amounts (even up to 12%) to rejuvenate AMs with high contents of RAP [27,30,33]. The presented results indicate that it is possible to effectively use an organic rejuvenator. The application of 100% RAP together with the rejuvenator Asfix Alfa allows for the creation of AMs with similar physical and mechanical parameters (in lowered technological temperatures) to those obtained using other technologies with high contents of RAP [19,20,21,22].

The initial fatigue tests for AMs with 20%, 40%, and 100% RAP contents using the organic rejuvenator indicate that incorporating a high content of RAP into AM did not lead to a significant difference in the fatigue resistance, which may suggest that the blending of the RAP and the virgin binder is working as well as with the virgin binder alone.

## 7. Conclusions

On the basis of the lab tests of two AMs of various RAP contents (20% and 40%), as well as tests done on trial road sections, including AM-100%, the following conclusions can be drawn:The rejuvenator used in the tested AMs improved the homogenization of RAP with the binder and virgin aggregate. Moreover, it has been proved that it is possible to use 20%, 40%, and even 100% contents of RAP in the mixtures thanks to the use of a rejuvenator that creates conditions for the effective homogenization of AM components.The use of the rejuvenator allowed the optimal use of RAP and obtained the correct content of voids in the pavement. What is more, lowering the compaction temperature by 20 °C in relation to the standard temperature used in HMA technologies makes it possible to use fluxed asphalt containing the rejuvenator without loss in any water or frost resistance.As the amount of rejuvenator used in a mixture increases, the quantity of voids drops in comparison with the samples where the agent was not used. The RAP sample without rejuvenator content presented an excessive content of voids: 9.8% at a compaction index of 99.7%.The lab designed mixtures AM-20% and AM-40% are characterized by twin density and bulk density as well as a similar content of voids. However, when the RAP content is 20%, the number of voids is bigger by 0.7% than in the case of the 40% RAP content in the AM. This is indirectly caused by an increased number of fine fractions in RAP which better fill in voids.A more advantageous test result was obtained for the designed AM-20% in rutting tests than for the AM-40%. The mixture with a greater content of RAP showed a slightly bigger proportional rut depth, which can prove the impact of the rejuvenator and its stabilization in the case of stiff mixtures.Very good water and frost resistance (ITSR) was obtained for both the AM-20% AM-and 40%, amounting to 92% and 93%, respectively. Verification tests concerning the checks of mixtures with various RAP contents showed very good, certified technical parameters, which guarantee the expected durability of the pavement.It has been established that the most important factors regarding the design of AMs with RAP are: the selection of the proportion of recycled material, the classification of the RAP fraction, and the determination of the proper amount of the rejuvenator in the mixture.Using RAP with an organic rejuvenator together lowered the technological temperature required and is the main factors consistent with the idea of balanced, pro-ecological road construction engineering.

## Figures and Tables

**Figure 1 materials-14-02567-f001:**
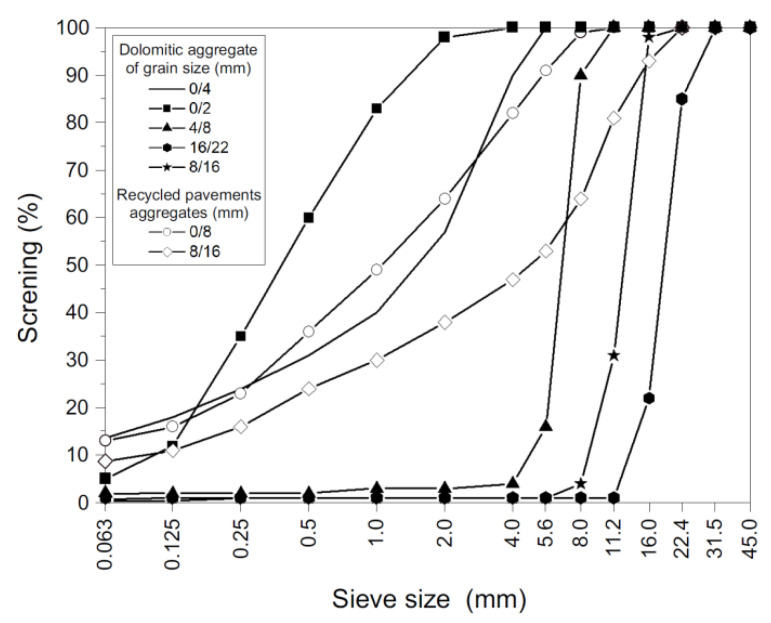
Grain size curves for the applied aggregates.

**Figure 2 materials-14-02567-f002:**
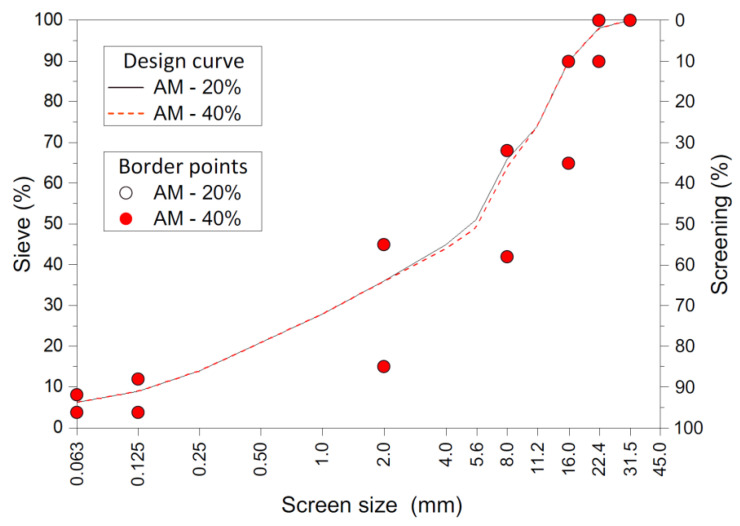
Grain size curves for AM-20% and AM-40%.

**Figure 3 materials-14-02567-f003:**
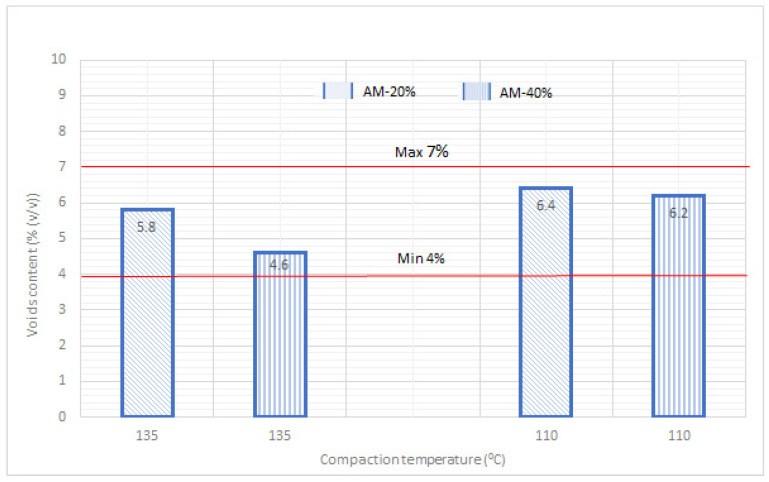
Test results of voids content for AM-20% and AM-40%.

**Figure 4 materials-14-02567-f004:**
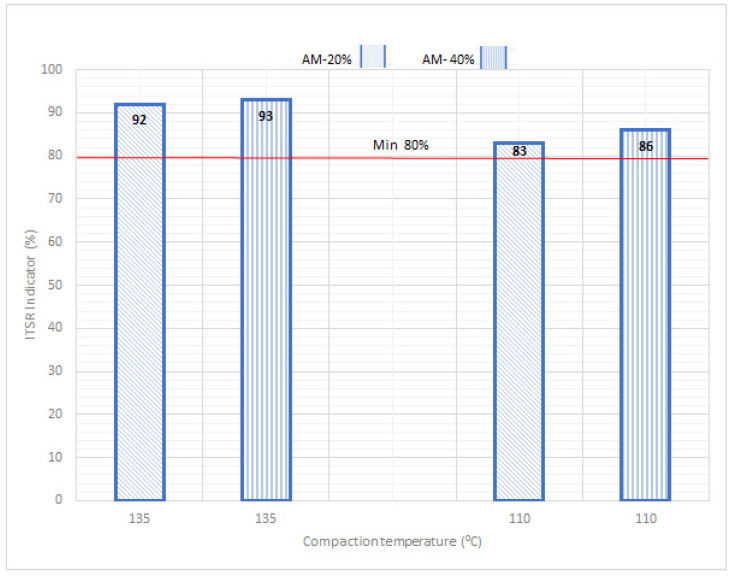
ITSR results for AM-20% and AM-40%.

**Figure 5 materials-14-02567-f005:**
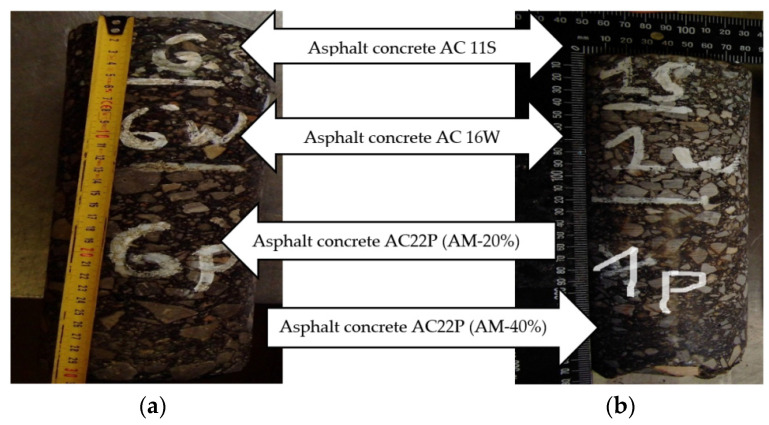
Extracted pavement cores: (**a**) AM-20% and (**b**) AM-40%.

**Figure 6 materials-14-02567-f006:**
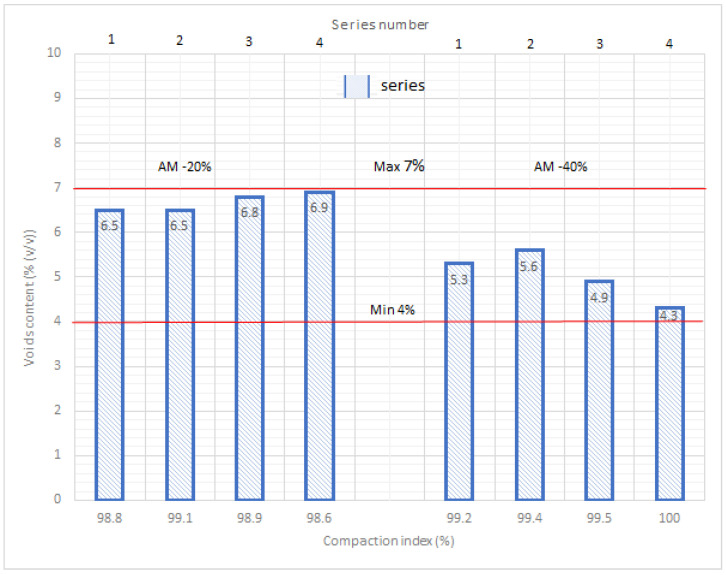
Lab tests results for samples from trial sections of AM-20% and AM-40%.

**Figure 7 materials-14-02567-f007:**
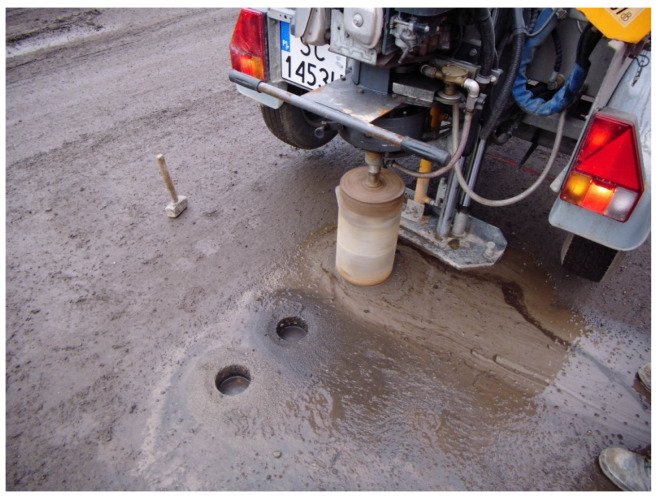
The drilling of the samples from the tested trial road pavement (AM-100%).

**Figure 8 materials-14-02567-f008:**
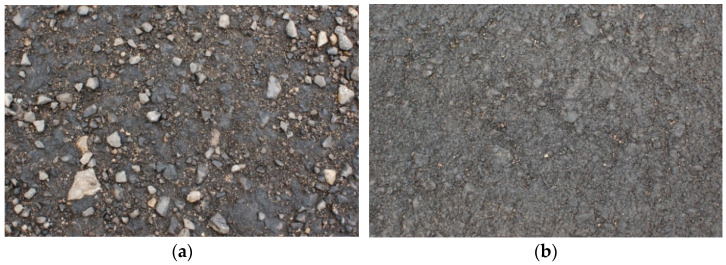
Trial road sections: (**a**) “gravel”—cold rolled destructed asphalt material and (**b**) hot, recycled in 120 °C, RAP of the same material.

**Figure 9 materials-14-02567-f009:**
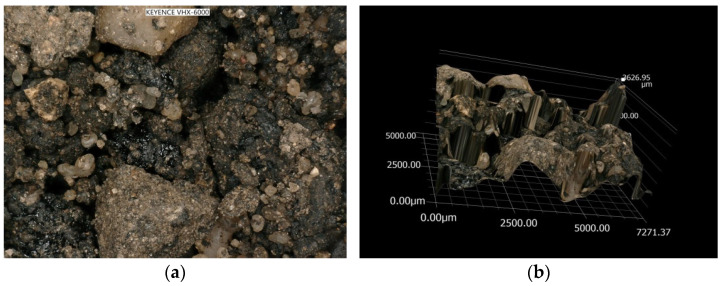
Microscopic image of granulate of two fractions: (**a**) surface image with visible grains and (**b**) spatial image of the RAP sample surface.

**Figure 10 materials-14-02567-f010:**
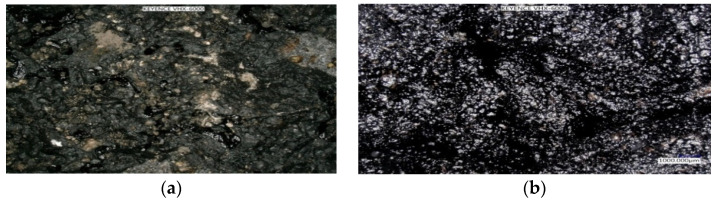
Granulate-based pavement surface in microscopic image (**a**) without rejuvenator—image of old, non-uniform structure and (**b**) with 1.3% of rejuvenator—image of rejuvenated, uniform structure.

**Figure 11 materials-14-02567-f011:**
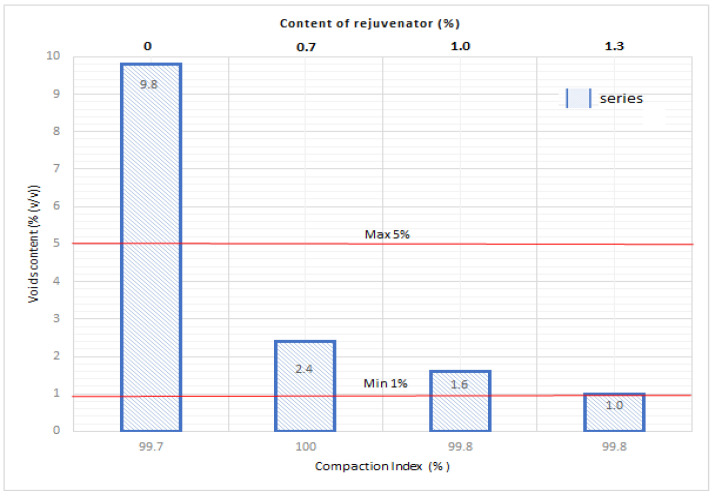
Voids content and compaction index of AM-100% in relation to the rejuvenator content.

**Table 1 materials-14-02567-t001:** The basic properties of various rejuvenators [46,47].

Properties	Characteristics		
	Asfix Alfa	Wetfix BE	Wetfix AP17
Color	light yellow	brown	brown
Consistence	liquid	liquid	liquid
Flash point (°C)	>210	>218	>240
Density in 15 °C (g/cm^3^)	0.918–0.925	0.88–0.98	0.90–0.99
Acid value (mg/g KOH)	85.5	<10	<10
Total base number (mg/g KOH)	<2.5	-	-
Pour point (°C)	−6	0	−7
Viscosity mPas	3100	<3000	<1000

**Table 2 materials-14-02567-t002:** Properties of binders used in mixtures.

Binder Types	Binder Properties	
	Penetration in 25 °C *	Softening Point (°C)
35/50	48	54
Binder reclaimed from RAP	21	65

* Unitless number corresponds with 0.1 mm immersion of the penetrating needle.

**Table 3 materials-14-02567-t003:** List of designed compositions of the AMs.

Components	Mixture Type	
	AM-20%	AM-40%
Limestone filler (%)	2.4	1.0
Dolomite aggregate of continuous grain size 0/4 mm (%)	17	6.5
Fine dolomite aggregate 0/2 mm (%)	12.5	9.6
Coarse dolomite aggregate 4/8 mm (%)	16.2	13
Coarse dolomite aggregate 16/22.4 mm (%)	11	10
Coarse dolomite aggregate 8/16 mm (%)	17	16
Recycled pavement aggregates 0/8 mm (%)	10	20
Recycled pavement aggregates 8/16 mm (%)	10	20
Total asphalt (%)	3.9	3.9

**Table 4 materials-14-02567-t004:** Binder content in the AMs.

Content of RAP in AM (%)	Content of Various Fractions of RAP in AM (%)	Content of Asphalt in RAP (%)	Content of Virgin Binder in Total Amount of Asphalt in AM (%)	Content of Binder from RAP in Total Amount of Asphalt in AM (%)
20	0/8 mm–10	5.8	1.9	2
	8/16 mm–10	4.3		
40	0/8 mm–20	5.8	2.9	1
	8/16 mm–20	4.3		

**Table 5 materials-14-02567-t005:** Volumetric test results of AMs.

Parameter Measured	Mixture Type		Requirements [48] for Traffic Category	
	AM-20%	AM-40%	KR1–2 *	KR3–4 *
Density of mineral skeleton (Mg/m^3^)	2.765	2.739	-	-
AM density (Mg/m^3^)	2.602	2.58	-	-
AM bulk density (Mg/m^3^)	2.45	2.465	-	-
Voids content (%)	5.8	4.5	4–8	4–7
Voids filled in with a binder (%)	61	67	50–74	-
Mineral mixture voids content (%)	14.8	13.5	≥14	-

* KR1–2 and KR3–4 denote respectively light traffic and medium/medium-heavy traffic [48].

**Table 6 materials-14-02567-t006:** The test results of the AMs resistance to durable deformations.

Parameter	Mixture Type		Requirements for Traffic Category	
	AM-20%	AM-40%	KR1–2	KR3–4
Rutting graph slope WTS_AIR_ (mm/10,000 cycles)	0.07	0.07	no requirements	≤0.30
Proportional rut depth PRD_AIR_ (%)	3.8	4.6	no requirements	≤9.0

**Table 7 materials-14-02567-t007:** The lab test results of AM composition of 100% granulate (average from four series taken from the test pavement).

Sieve Size (mm)	Sieving (%) of Tested Material under Extraction with Separation of Mineral Fractions	Deviations Obtained in Comparison to Basic Recipe Showing Dosing Accuracy in Site Conditions	Acceptable Deviations in Accordance with PN-EN 13108-21:2008 [63]
16	100	0	-
11.2	92	+2	/−8, +5/
8.0	82	−3	-
5.6	72	+2	±7
4.0	63	+1	-
2.0	50	−5	±6
1.0	39	+7	-
0.5	30	+3	-
0.25	21	+1	-
0.125	15	+1	±4
0.063	11.7	+0.3	±2

## Data Availability

The data presented in this study are available within the article.

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
