# Peer review of "Influence of a Rejuvenator on Homogenization of an Asphalt Mixture with Increased Content of Reclaimed Asphalt Pavement in Lowered Technological Temperatures"

_materials, 2021, doi:10.3390/ma14102567_

Round 1

Reviewer 1 Report

The manuscript describes an application study on warm asphalt recycling, with different proportions of reclaimed asphalt, and using a commercial rejuvenator. The manuscript describes the materials and field trial, however it is very limited in terms of the analysis and discussion of a research paper. In addition, the terminology used throughout the manuscript is not pavement engineering. For all of these, I recommend the manuscript to be rejected, with the possibility of resubmitting if significantly improved.

The authors should consider the following:  

- The terminology of pavement engineering is frequently not followed, for example: reclaimed asphalt instead of the asphalt granulate; asphalt, asphalt mixture or bituminous mixture instead of the mineral-asphalt mixture (do the authors know any bituminous mixture with non-mineral structure??); dust or filler instead of the flour….

- Why is the rejuvenator referred to as an adhesive agent? What is the effect of the rejuvenator on the rheology of the binder? This should be studied.

- The test methods used need to be described. The performance characterization of the materials is not sufficient to evaluate the technique (fatigue resistance is essential).     

Reviewer 2 Report

Title:

It is necessary to specify the type of recycled aggregate used in the work. Modify the title

Abtract:

It is necessary to make evident the need to carry out this research (it should be included at the beginning of this section).

It is necessary that the most significant results of this research be stated in quantitative terms.

Keywords:

Include recycled pavement aggregates.

Introduction:

It is necessary to perform a search of the last 10 in databases such as Compendex and Scopus or (WoS).

It is recommended to include precise data on the properties and variables mentioned in this section.

Materials:

It is necessary that the authors focus on providing accurate data on the tests and materials used (avoid redundancy and digression; the authors are not writing a technical book, it is a scientific article).

It is necessary to include precise and accurate information on the "bonding agent" (brand name, or failing that, chemical composition), other researchers should be able to replicate this work; the above applies to all components or materials used).

Table 1. You must be immediately after the paragraph that refers to it in the text.

Description Contents....:

Translated with www.DeepL.com/Translator (free version)

It is recommended that Table 2 be replaced with a graph.

Table 3. It is necessary to justify the selected variables.

Figure 1. Verify that all the sentences can be read.

Methodology of the investigation:

It is necessary to indicate make, model and country of manufacture of all laboratory equipment used.

Test results:

The use of alternative tests is recommended to corroborate or explain the exposed behavior.

Tests on road sections:

Specify the location of the test sections.

Table 8. It is recommended that the results be presented graphically (delete table).

Figures 5 and 6 are considered not to provide relevant information for the work (delete them).

Discussion:

A more critical and deductive (not only narrative) analysis of the results is needed.

The results of this research need to be contrasted with works of other researchers.

Conclusions:

Revise after making the requested modifications.

Bibliography:

Is limited or reduced, see previous comments.

Reviewer 3 Report

The paper provides laboratory experiments on HMA made of recycled materials in different contents. The strength of the paper is backing up the experimental results with the observations and tests on the field trials. The manuscript is useful to the research and industry community as it promotes the application of recycled materials up to 100% in the production of asphalt. However, the quality of the texts needs significant improvement to be suitable for publication. The literature review does not have a flow and it is difficult to understand the connection between the paragraphs. Hence, a complete proofread and improvement of the manuscript is highly recommended. Also, more correspondence between the undertaken research and the review of the literature is expected. The following recommendations are meant to assist the authors in improving their manuscript:

  • The points in the first paragraph are expected to be supported by citing relevant references.
  • The sentence starting in line 66 needs a citation. In the following sentence, what is “The article”? is it the article from which the previous sentence was extracted?
  • Line 72: “big” is not an appropriate term. Please replace with a more objective term!
  • Line 75: More description and examples of “the recent technologies” should be provided and the references should be cited.
  • Line 80: Rather than writing as “the paper” or “the article” followed by the reference number, it is suggested to write the authors’ names and publication date followed by the reference number. E.g. Huang et al. (2005) [10] presented …
  • Same comment as above for reference 11 (line 86). Also, the introduction section jumps from this article to the other without any connection between the results from the literature.
  • Line 88: Hot Mix Asphalt (HMA) is a more common term. Please amend throughout the manuscript.
  • Lines 166-171: more explanation on the natural origin of the agent should be provided. It is not clear what the origin of this agent is!
  • Table 1: Characteristics of the adhesive agent used in this research is recommended to be compared with traditional chemical agents, by adding 1-2 columns to Table 1 and providing characteristics of traditional chemical agents. Also, what is the source for the info in Table 1? Also, Tables and Figures are recommended to be placed after the paragraph in which they are referred to (not before it).
  • Table 3: how is recycled “aggregate” a type of binder?!
  • Line 332: Please replace “fresh (new)” with virgin, which is a more appropriate term.
  • What do the authors mean by “ecological agent” in the manuscript? DO they mean “organic agents”?

Round 2

Reviewer 1 Report

The manuscript was sufficiently improved to recommend accepting it.

Reviewer 2 Report

Dear authors:

Many improvements have been made in this new version of your work. However, there is one that still persists:

Question

Test results:

The use of alternative tests is recommended to corroborate or explain the exposed behavior.

Re: Thank you for remark. All conducted tests were performed in an accredited laboratory Labor Aquila in accordance with the highest research standards. Accreditation for the road laboratory enables pan-European testing. Moreover, some tests are verified by other laboratories to confirm the quality of results.

New argument: It is possible that I did not explain well. I do not question the quality of the laboratories and equipment used in the tests; what I request is that the authors carry out new alternative tests that allow us to corroborate that those carried out are valid. For example, chemical characterization techniques, such as XDR, or tests that demonstrate the structure of the matrix, such as SEM

Reviewer 3 Report

The authors have significantly improved the manuscript. Although I could not find a document with responses to my comments, the requested modifications have been implemented in V2.
